# Seasonal Variation of Shoreline Fish Assemblages at Two Stations in the Southern Branch of the Yangtze River Estuary

**DOI:** 10.3390/biology14121785

**Published:** 2025-12-14

**Authors:** Bo Feng, Guangpeng Feng, Xuzhe Gu, Ju Yang, Qingbo Zhang

**Affiliations:** 1Shanghai Yangtze Estuary Fishery Resources Proliferation and Ecological Restoration Engineering Technology Research Center, East China Sea Fisheries Research Institute, Chinese Academy of Fishery Sciences, Shanghai 200090, China; fb991221@163.com (B.F.); m13256706751@163.com (X.G.); 13214472434@163.com (J.Y.); zhangyuf4@163.com (Q.Z.); 2College of Fisheries and Life Sciences, Dalian Ocean University, Dalian 116023, China

**Keywords:** fish community structure, seasonal dynamics, ABC curves, species diversity, Yangtze River Estuary

## Abstract

We conducted four seasonal shoreline surveys in 2024 at two sites in the Yangtze River Estuary using the same gear and timing. We recorded 47 species. A few common fishes dominated the catch, while many others were rare. Most species lived near the bottom or in mid-water and fed mainly on animals or mixed diets. Species composition changed with the seasons: spring and summer were most similar, and winter was the most different. ABC curves based on relative abundance and biomass indicated moderately disturbed assemblages in spring and summer, but less disturbed communities in autumn and especially winter (W from −0.066 to 0.276). These findings provide a clear, repeatable baseline for tracking estuary health, guiding habitat restoration, and supporting protection of threatened species such as the Chinese sturgeon (*Acipenser sinensis*).

## 1. Introduction

The Yangtze River Estuary, located at the confluence of the Yangtze River, the East China Sea, and the Yellow Sea, is the largest and most complex estuarine ecosystem in the western Pacific Ocean [1,2], Driven by the combined influence of the Yangtze River’s freshwater discharge, the Taiwan Warm Current, the Yellow Sea cold water mass and coastal waters from northern Jiangsu, the region exhibits highly dynamic hydrological and physicochemical gradients [3]. These conditions create a nutrient-enriched environment that supports high primary productivity and sustains diverse fish communities and other fish assemblages. The estuary functions as an important breeding, nursery, and feeding ground for numerous fish species [4], and its complex food-web interactions are essential for maintaining biodiversity and ecosystem stability [5]. Given its ecological and economic significance, the Yangtze River Estuary represents a vital ecological corridor linking freshwater and marine ecosystems.

*Acipenser sinensis*, a critically endangered anadromous fish endemic to China, relies on the Yangtze River and adjacent estuarine waters during key life-history stages [6]. Designated as a national first-class protected fish species in 1988 and listed as “Critically Endangered” by the IUCN in 2010 [7], its population has dramatically declined due to overfishing, pollution and habitat fragmentation [8,9]. The construction of large dams has further interrupted migratory routes and reduced connectivity between spawning and feeding grounds [10]. Juvenile *A. sinensis* inhabit shallow brackish waters around Chongming Island and adjacent tidal flats during their transition from freshwater to marine environments. Previous studies have shown that they mainly forage in intertidal and nearshore habitats with sandy or sandy–muddy substrates, where small demersal and benthic fishes, together with some benthic invertebrates, constitute the bulk of their diet [11,12,13,14]. The survival and recruitment success of juveniles are therefore closely linked to the availability of such prey resources and the maintenance of suitable nearshore feeding habitats [15]. Along the southern branch of the estuary, shallow intertidal and nearshore zones along revetments and tidal flats form an important component of the feeding habitats used by juvenile *A. sinensis*; therefore, describing the composition and seasonal dynamics of co-occurring shoreline fish assemblages provides essential background on the temporal availability of their potential prey within this habitat corridor and helps to establish ecological baselines relevant to the conservation and restoration of *A. sinensis* habitats.

Fish communities are widely recognized as sensitive indicators of estuarine ecosystem health because their composition reflects integrated responses to both biotic interactions and abiotic variability [16,17,18]. Seasonal fluctuations in hydrological and physicochemical factors—such as temperature, salinity, and dissolved oxygen-can significantly alter community structure and abundance [2]. These seasonal changes also affect the availability of small demersal and benthic fishes that form important prey for juvenile *A. sinensis* [11]. However, despite the ecological importance of the Yangtze River Estuary, few studies have systematically documented the seasonal dynamics and turnover of intertidal fish assemblages dominated by fishes in this region. Existing research in the area has primarily focused on sturgeon migration and physiology, whereas long-term observations of intertidal fish-community structure remain scarce. A detailed understanding of fish assemblage composition and its seasonal variability is therefore essential for evaluating estuarine ecological status and for providing a reference baseline for ongoing restoration programs [19,20].

This study therefore had three main objectives: (1) to document the taxonomic composition and ecological guild structure of shoreline fish assemblages in the southern branch of the Yangtze River Estuary; (2) to quantify their seasonal dynamics in terms of species richness, diversity and dominance across four sampling periods; and (3) to characterize inter-seasonal changes in community structure and similarity among seasons. Establishing these community-level patterns provides a standardized seasonal baseline for shoreline fish assemblages along this engineered estuarine sector and offers ecological background for future assessments of nursery and feeding habitats of juvenile *A. sinensis*.

## 2. Materials and Methods

### 2.1. Survey Sites and Fishing Gears

Seasonal surveys of intertidal fish assemblages were conducted in the Yangtze River Estuary from May to December 2024, covering spring, summer, autumn and winter. Fish were sampled at two intertidal shoreline sites along the southern branch (Figure 1): (1) the Baozhen waters, located at the western end of Chongming Island, and (2) the Dongtan waters at the southern outlet of the estuary. At each site, one 300 m fixed stake net was deployed, flanked by two 100 m triple-layer gillnets (1.5 m in height, 2.0 cm minimum mesh size) to ensure consistent spatial coverage across the estuarine channel. Both the fixed stake net and the triple-layer gillnets were fully bottom-set in shallow nearshore waters and fixed to the shoreline, so that they consistently fished the near-bottom zone in each season. The triple-layer gillnets consisted of a finer-mesh inner panel sandwiched between two larger-mesh outer panels, providing size-selective sampling of small to medium-sized fishes. Nets were set at sunset and retrieved after approximately 12 h at sunrise to capture crepuscular and nocturnal activity. The tidal stage at each deployment and retrieval was recorded, and setting/hauling were conducted at comparable tidal phases in each season. During our surveys, water depth at the sampling locations generally ranged from approximately 0.3 to 1.5 m during net deployment and retrieval. In this study, the “intertidal zone” refers to the shallow shoreline waters between the mean low and high tide levels along artificial revetments and gently sloping beaches, which are alternately exposed and inundated by semi-diurnal tides. A field photograph of the fixed stake net and the triple-layer gillnets used in this study has been added (Figure 2) to illustrate their configuration and deployment.

The two shoreline sites were selected based on several criteria. First, both sites are located along the southern branch of the Yangtze River Estuary within the known habitat range of juvenile *A. sinensis*, where previous surveys and by-catch records have documented the occurrence of juveniles. Second, the sites represent typical nearshore waters of the southern branch, with shallow brackish waters and soft substrates that provide potential feeding habitats for demersal fishes. Both sites lie along the mid-lower reach of the southern branch under similar hydrological and salinity regimes, and together they typify nearshore waters of the southern branch habitats of this sector. under similar hydrological and salinity regimes, with nearshore depths generally less than 2 m during fishing and predominantly sandy–muddy substrates and are situated outside the main navigation channel. Third, the locations offered safe and repeated access for seasonal sampling while avoiding major navigation channels and intense ship traffic. Together, these criteria ensured that the surveys captured intertidal fish assemblages in areas that are ecologically relevant to juvenile *A. sinensis* and logistically feasible for field sampling.

All captured fishes were transported to the laboratory immediately after capture for subsequent processing.). Species identification followed *Fish of the Yangtze Estuary* (2nd Ed.) [21]. For each specimen, total length (TL, cm) and wet weight (W, g) were measured to characterize body size and to provide biomass data for calculating the index of relative importance of dominant species. These biological data formed the basis for subsequent analyses of species composition, community diversity, and abundance patterns. The surveys were conducted under institutional approval and in compliance with animal welfare guidelines.

### 2.2. Data Analysis

For all fish assemblage analyses, catches from Baozhen and Dongtan were pooled by season to derive an integrated seasonal baseline of nearshore waters fish assemblages for this sector of the southern branch. Given the single net-set per station per season and the broadly similar habitat and gear configurations at the two sites, formal between-station comparisons were not attempted. All statistical analyses and figure generation were conducted in R (v4.4.3; R Foundation for Statistical Computing, Vienna, Austria).

#### 2.2.1. Biodiversity Index

In this study, the biodiversity of the intertidal fish assemblages was quantified using four standard indices: Margalef’s richness index (*D*), Shannon-Wiener diversity index (*H*′), Pielou’s evenness index (*J*′) and Simpson diversity index (*C*). Their formulas are as follows:(1)$$D=\frac{S-1}{lnN}$$(2)$$H’=-\sum_{i=1}^{S}PilnPi$$(3)$$J^{\prime }=\frac{H^{\prime }}{lnS}$$(4)$$C=1-\sum_{i=1}^{S}{Pi}^{2}$$

In Formulas (1)–(4), *S* denotes the total number of fish species recorded in all samples, and *N* represents the total number of individuals of all fish assemblages across those samples, and Pi is the relative abundance of the ith species, calculated as *n*i/*N*, where *n*i is the count of individuals of species i.

#### 2.2.2. Ecological Guild Classification

Fish in the estuarine zone can be classified into four ecological guilds-freshwater, estuarine-resident, migratory, and marine species-based on their tolerance to salinity and estuarine conditions [22]. They also fall into two behavioral categories, migratory and sedentary, according to movement patterns; into four trophic groups-omnivores, carnivores, filter feeders, and herbivores-according to feeding habits; and into three spatial strata-pelagic, midwater, and benthic-according to preferred depth zones. These guild definitions were primarily developed for fishes, which constituted the majority of the assemblage, but the overall community analyses considered all captured fish assemblages.

#### 2.2.3. Dominant Species

The relative importance index (*IRI*) was employed to identify dominant fish assemblages and quantify each species’ contribution to community structure. *IRI* is calculated as:*IRI* = (*N*% + *W*%) × *F*%(5)

In Equation (5), where *N*%, *W*% and *F*% are the percentage of numerical abundance, biomass and frequency of occurrence of species *i*, respectively. Species are classified by *IRI* value as follows: classifies species as dominant (*IRI* ≥ 1000), important (100 ≤ *IRI* < 1000), common (10 ≤ *IRI* < 100), or rare (*IRI* < 10), thereby facilitating a concise assessment of their ecological roles within the community.

#### 2.2.4. Composition Similarity

The similarity between fish communities at different survey sites was quantified using Jaccard’s similarity index [23]. defined as:*Cj* = *c*/(*a* + *b* − *c*) × 100%(6)

In Formula (6), *a* and *b* denote the numbers of species in the fish communities of the first and second seasons, respectively, and c is the number of species shared by both seasons. The Jaccard index (*Cj*) ranges from 0 to 1, with higher values indicating greater compositional similarity. Following [24], Values of *Cj* are interpreted as follows: 0–0.25 indicates extremely low similarity, 0.25–0.50 low similarity, 0.50–0.75 moderate similarity, and 0.75–1.00 high similarity.

#### 2.2.5. Community Seasonal Alternation and Migration Indices

The community seasonal alternation index (AI) and migration index (MI) were computed as follows to quantify temporal turnover and species movement between seasons:(7)$$\mathrm{AI}\;=\;\frac{C+B}{A-R}\;\times \;100$$
(8)$$\mathrm{MI}=\frac{C-B}{A-R}\;\times \;100$$

In Equations (7) and (8), *A* represents the total number of species observed in a given season, *C* is the count of species newly immigrating into the community that season, *B* is the count of species emigrating out, and *R* is the number of species persisting across all four seasonal surveys. The alternation index (AI) quantifies the rate of species turnover-and hence loss of community stability such that lower AI values correspond to greater stability. The migration index (MI) measures the balance between incoming and outgoing species: a positive Ml (*C* > *B*) indicates net immigration, whereas MI near zero (*C* ≈ *B*) signifies a dynamic equilibrium in species movement [25].

#### 2.2.6. Abundance Biomass

The abundance-biomass comparison (ABC) method, grounded in r- and K-selection theory [26], was used to assess the seasonal disturbance status of the fish assemblages. In this approach, cumulative dominance curves of biomass and abundance are plotted on the same axes. The relative position of the two curves, together with the value of the W statistic, indicates the degree of disturbance. When the biomass curve lies above the abundance curve, W is positive and the community is interpreted as relatively undisturbed and K-selected. When the biomass curve lies below the abundance curve, W is negative and the community is considered to be strongly disturbed and more r-selected. The W statistic is calculated as:(9)$$W=\sum_{i=1}^{S}\frac{\left(Bi-Ai\right)}{50(S-1)}$$

In Formula (9), *S* denotes the total number of species, While *Bi* and *Ai* represent, respectively, the cumulative percentages of biomass and of abundance up to the *i*th species on the ABC curve.

## 3. Results

### 3.1. Species Composition and Ecological Guilds of Intertidal Fish Assemblages

This study collected a total of 47 fish species comprising 3310 individuals across four seasonal surveys (spring, summer, autumn and winter) in 2024. These species belonged to 37 genera, 18 families and 10 orders (Table 1). The most speciose order was Cypriniformes, with 19 recorded species, followed by Perciformes with 10 species, and Siluriformes and Acipenseriformes with two species each. The family Cyprinidae contributed the highest proportion of species, with 19 species accounting for 40.43% of the total species richness. In addition, one individual of the endangered *A. sinensis* was recorded.

Based on ecological guild classification, carnivorous fishes were the most numerous, with 27 species (57.45% of all species), followed by 13 omnivorous species (27.66%), and 5 filter-feeding and 2 herbivorous species. In terms of vertical habitat use, 22 benthic (bottom-dwelling) species (46.81%) were recorded, while 13 species inhabited the upper-mid water column and 12 species occurred predominantly in the mid-lower water column. The most frequently encountered species during the monitoring period were *Coilia nasus*, *Parabramis pekinensis*, *Cynoglossus gracilis*, *Culter alburnus*, *Lateolabrax japonicus* and *Liza haematocheila* (Figure 3).

### 3.2. Analysis of Dominant Fish Species

During the four seasonal surveys in 2024, the composition of dominant fish species showed pronounced seasonal variation (Table 2). The number of dominant species was five in spring, seven in summer, six in autumn and five in winter. The number of important species (i.e., species with relatively high *IRI* values) was highest in spring (13 species), decreasing to 10 in summer, 9 in autumn and only 3 in winter. In spring, *C. nasus* had the highest index of relative importance (*IRI* = 6365.79), followed by *P. pekinensis* (*IRI* = 2509.13). In summer, *C. nasus* remained the most dominant species (*IRI* = 3005.06), with *L. haematocheila* ranking second. In autumn, *L. haematocheila* (*IRI* = 3320.97) and *C. nasus* (*IRI* = 3229.19) jointly dominated the assemblage. In winter, *L. haematocheila* (*IRI* = 7666.47) and *L. japonicus* (*IRI* = 4855.04) exhibited the highest dominance. *C. gracilis* remained a dominant species in all four seasons, whereas *C. nasus* and *L. japonicus* were dominant in three of the four seasons.

### 3.3. Intertidal Fish Assemblages

*H*′ ranged from 1.7970 to 2.4410 and was highest in autumn, with *C* showing a similar pattern (Figure 4). *D* was greatest in spring, indicating higher species richness during the warmer seasons, whereas *J*′ peaked in winter, reflecting a more even distribution of abundances in that season. Overall, these indices point to a moderately diverse fish assemblage throughout the year, with richer communities in spring–summer and more even, though less rich, assemblages in winter.

### 3.4. Seasonal Similarity of the Fish Assemblages

The Jaccard similarity index (*Cj*) between seasons ranged from 17.14% to 50.00% (Table 3). Spring and summer shared the highest similarity (*Cj* = 50.00%), whereas spring and winter were the most distinct (*Cj* = 17.14%); other seasonal pairs showed intermediate overlap (~30%).

### 3.5. Seasonal Changes in Fish Assemblages

The alternation and migration indices showed marked seasonal variation (Figure 5). The migration index was positive only in spring, indicating more species immigrating than emigrating, whereas it was negative in summer, autumn and winter, consistent with a more stable assemblage during these seasons. Among the four seasons, immigration was greatest in spring and emigration was highest in summer.

### 3.6. Abundance-Biomass Comparison (ABC) Curves

The ABC curves (Figure 6) and W statistics showed slightly negative values in spring and summer (W = −0.066 and −0.018, respectively) and positive values in autumn and winter (W = 0.043 and 0.276). This pattern suggests relatively higher disturbance in spring–summer and comparatively lower disturbance in autumn–winter.

## 4. Discussion

### 4.1. Changes in Species Composition and Ecological Types

The Yangtze River Estuary exhibits pronounced seasonal variation in water temperature, which can influence community structure by altering fish species composition [27]. In this study, 34, 40, 26, and 8 fish species were recorded in spring, summer, autumn, and winter, respectively. This pattern is consistent with the subtropical monsoon climate, in which warmer summer temperatures enhance activity, growth and recruitment of estuarine fishes, whereas cooler winter conditions restrict their occurrence and catchability [28]. Similar seasonal effects of temperature and salinity on estuarine fish assemblages have also been documented in other systems, such as Chesapeake Bay [29].

Dominant taxa such as *L. haematocheila*, *C. nasus*, and *L. japonicus*-important prey species for *A. sinensis* [15]—were most abundant during summer and autumn. Their seasonal increase likely enhances potential food availability for juvenile sturgeons, which depend on the estuary as a feeding and acclimation habitat before migrating to the sea [14]. The consistent occurrence of *C. gracilis* across all seasons also contributes to a relatively stable prey base, maintaining trophic continuity in the estuarine ecosystem. In addition to these dominant fishes, a single individual of *A. sinensis* was captured during the year-long monitoring, which was originally designed to survey juvenile sturgeons along the southern branch. This record confirms that juvenile sturgeons still use these nearshore waters during their downstream migration, even though the detection rate in our survey was very low. The coexistence of abundant demersal fishes in nearshore habitat and occasional occurrences of *A. sinensis* is therefore consistent with previous descriptions of sturgeon feeding grounds in this region. However, in the context of the long-term decline of wild *A. sinensis* and the increasing reliance on hatchery releases in the Yangtze River system, a low encounter rate at a few stations over one year is more likely to reflect the generally low local density and survival of juveniles during our sampling window than to indicate that this area makes little contribution to the population. Given the limited spatial coverage and sampling frequency of the present survey, our data are not sufficient to quantify the relative contribution of this particular sector to the overall population, and more targeted, multi-site and multi-year monitoring will be required to evaluate the importance of different nursery and feeding areas for juvenile *A. sinensis*.

### 4.2. Fish Diversity, Community Stability, and Potential Relevance to Habitat Assessments

Diversity indices revealed that the fish community was most diverse in autumn (*H*′ = 2.4410), with relatively higher species richness in spring and summer and higher evenness in autumn and winter (*J*′ up to 0.8642). Together, these patterns indicate a moderately diverse assemblage throughout the year, with comparatively richer communities in spring–summer and more even assemblages in autumn–winter. From a habitat perspective, periods with higher richness and/or evenness likely correspond to a broader and more continuous spectrum of potential prey resources for juvenile *A. sinensis*, although diet was not quantified in this study. The positive relationship between biodiversity and ecosystem stability is well documented; more diverse fish assemblages generally exhibit greater resistance and resilience to environmental fluctuations [30]. High biodiversity in estuarine systems is also linked to enhanced ecological functions, including nursery provision for endangered and commercially important species [20].

The ABC curve analysis [31] showed slightly negative W values in spring (W = −0.066) and summer (W = −0.018), with the biomass and abundance curves lying close together, indicating moderately disturbed assemblages dominated by smaller, short-lived species. In autumn, W became slightly positive (W = 0.043), and in winter it increased further (W = 0.276), with the biomass curve lying above the abundance curve, suggesting a shift towards less disturbed communities with a relatively greater contribution of larger and longer-lived taxa in the colder season [32]. Given the small number of sampling events (one net-set per station per season), these differences in W should be interpreted as descriptive patterns rather than definitive evidence of shifts in disturbance regime.

From a conservation perspective, the combination of high species richness in spring–summer, higher evenness in autumn–winter and the tendency towards lower disturbance in winter suggests that prey-field conditions for juvenile *A. sinensis* may be seasonally variable but generally favorable across much of the year, albeit with changing species composition and size structure. Although diet composition and sturgeon occurrence were not quantified in this study, the seasonal variation in nearshore waters fish assemblages documented here can therefore be used as ecological background when identifying priority seasons and habitats for protecting or enhancing nearshore feeding grounds of juvenile *A. sinensis*.

This study provides a site-based seasonal baseline for fish assemblages in the southern branch of the Yangtze River Estuary. Future monitoring should integrate environmental variables (e.g., temperature, salinity, dissolved oxygen, pH), expand spatial replication, and employ multivariate analyses (e.g., PERMANOVA/ANOSIM and ordination) to test seasonal and spatial hypotheses.

### 4.3. Study Significance and Limitations

Despite its limited spatial coverage, this study provides a standardized seasonal baseline description of intertidal fish assemblages at two core stations in the southern branch of the Yangtze River Estuary. By jointly analyzing diversity, dominance, similarity/turnover, and ABC, we identify sentinel taxa and practical monitoring windows—spring–summer for overall biodiversity status and autumn–winter for stress and dominance shifts. These results offer operational indicators and a comparable time-series framework that can be directly integrated with future environment-integrated monitoring and, where relevant, sturgeon-focused assessments.

This study has several limitations that should be considered when interpreting the results. First, the sampling design was restricted to two fixed nearshore waters sites along the southern branch of the Yangtze River Estuary. Although these sites were selected within the known habitat range of juvenile *A. sinensis*, they cannot capture the full spatial heterogeneity of intertidal and nearshore habitats throughout the estuary. Because samples from the two sites were pooled for seasonal analysis, potential small-scale differences between stations could not be evaluated., potential small-scale differences between stations could not be evaluated. Second, we used two types of nearshore fishing gear (a fixed stake net and a beach seine) that both operated in shallow nearshore waters; as a result, species that mainly inhabit deeper channels or avoid nearshore waters may still be under-represented in the catches. Third, environmental variables (e.g., salinity, turbidity, current velocity) were not measured concurrently with the biological surveys, which precludes direct modeling of environment-community relationships in this study. Despite these limitations, the present work provides one of the first seasonal descriptions of intertidal fish assemblages along nearshore waters of the southern branch and can serve as a useful baseline and reference point for the design of broader, multi-site and environment-integrated monitoring programs in the future.

Accordingly, we interpret our findings as baseline information on co-occurring nearshore waters fish assemblages and potential prey fields within the known habitat range of juvenile *A. sinensis*, rather than as a direct assessment of sturgeon habitat suitability, which will require future integrated fish–environment–sturgeon surveys.

## 5. Conclusions

The intertidal fish assemblages in the southern branch of the Yangtze River Estuary, within the known habitat range of *A. sinensis*, are dominated by cyprinid species and by sedentary, carnivorous, bottom-dwelling fishes. A small group of taxa, including *C*. *gracilis*, *C*. *nasus*, *L*. *japonicus*, *P*. *pekinensis*, *C*. *alburnus* and *L*. *haematocheila*, consistently structured the assemblages across seasons. Diversity metrics and ABC curves together indicate a moderately diverse community with clear seasonal shifts in richness, evenness and disturbance, with comparatively more disturbed conditions in spring–summer and more stable assemblages in autumn–winter. Overall, these site-based seasonal baselines provide a first standardized description of intertidal fish assemblages in nearshore waters of the southern branch and can inform the design of future expanded, environment-integrated monitoring and, where relevant, future assessments of juvenile *A. sinensis* feeding habitats.

## Figures and Tables

**Figure 1 biology-14-01785-f001:**
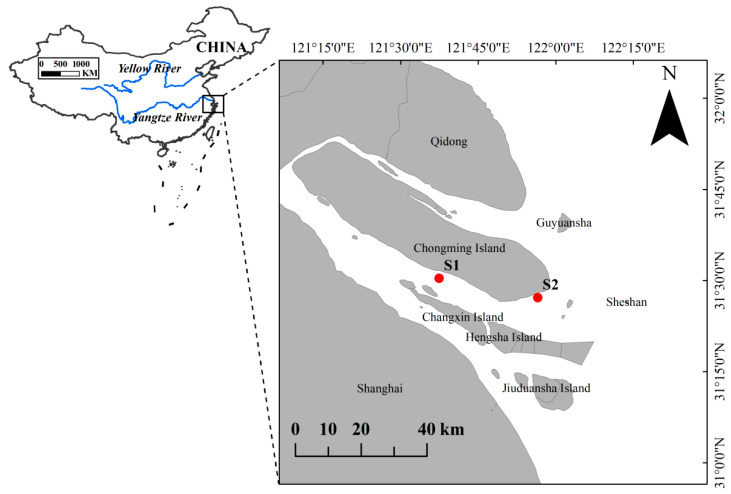
Study area and Sampling stations in the southern branch of the Yangtze River Estuary. Site 1 (121°38′ E, 31°31′ N), Site 2 (121°57′ E, 31°27′ N).

**Figure 2 biology-14-01785-f002:**
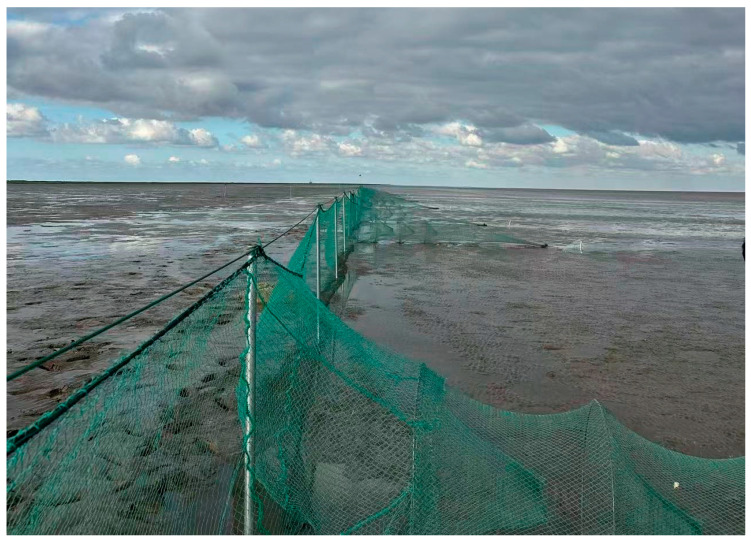
Sampling sites and Fishing gears in the southern branch of the Yangtze River Estuary.

**Figure 3 biology-14-01785-f003:**
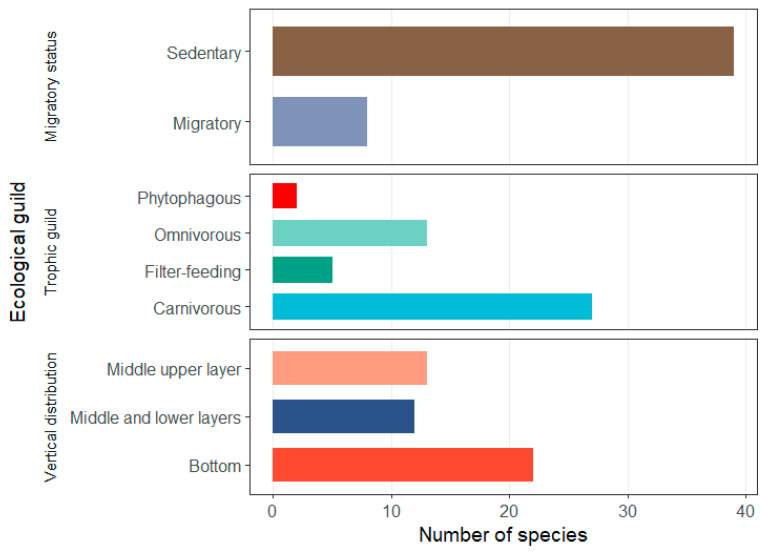
Ecological guild composition of intertidal Fish assemblages in the southern branch of the Yangtze River Estuary.

**Figure 4 biology-14-01785-f004:**
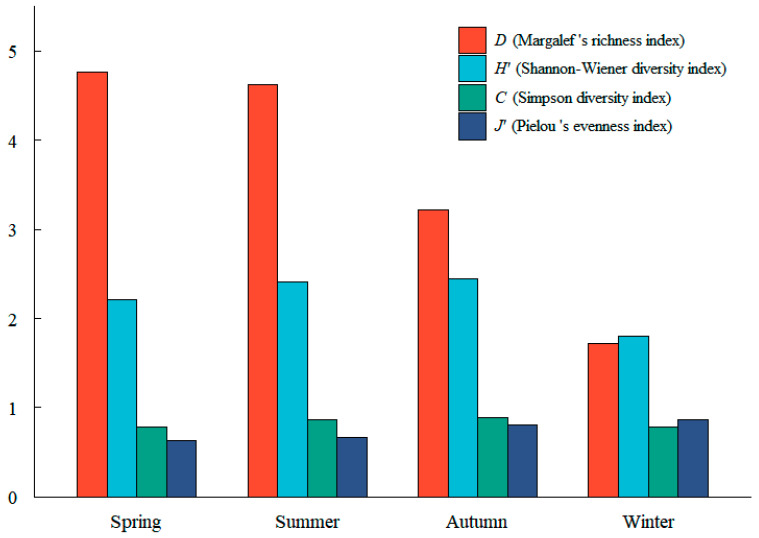
Seasonal variation in community diversity indices of intertidal Fish assemblages at two nearshore waters stations in the southern branch of the Yangtze River Estuary.

**Figure 5 biology-14-01785-f005:**
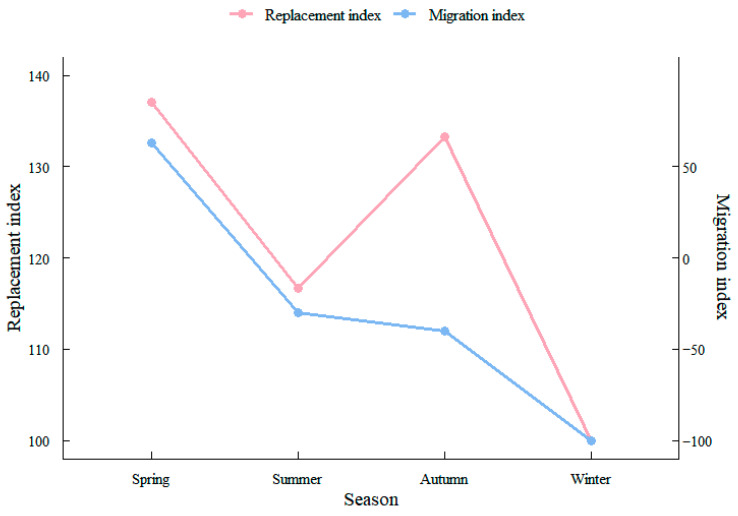
Seasonal variation in fish assemblages turnover index and migration index in the South Branch of the Yangtze River Estuary.

**Figure 6 biology-14-01785-f006:**
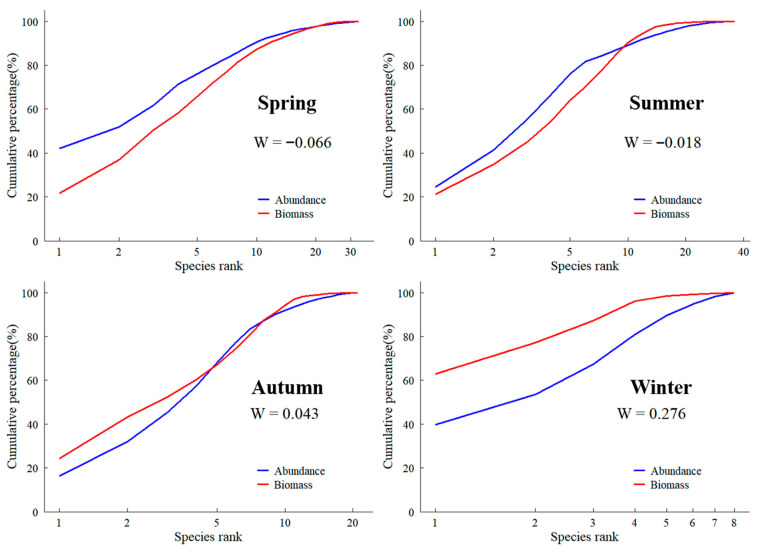
ABC curve of Fish assemblages in the South Branch of the Yangtze River Estuary in different seasons.

**Table 1 biology-14-01785-t001:** Composition and Spatial distribution of Fish assemblages in the South Branch of the Yangtze River Estuary.

Species	2024				Ecological Guild
	Spring	Summer	Autumn	Winter	
Cyperiniformes					
*Parabramis pekinensis*	80	327	58	0	Fr, S, C, U
*Culter alburnus*	38	271	32	0	Fr, S, C, U
*Saurogobio dumerili*	4	42	5	0	Fr, S, C, D
*Pseudobrama simoni*	22	6	2	0	Fr, S, O, L
*Cyprinus carpio*	6	21	0	0	Fr, S, O, L
*Hemiculter bleekeri*	17	3	0	0	Fr, S, O, U
*Squaliobarbus curriculus*	8	7	0	0	Fr, S, C, U
*Hypophthalmichthys molitrix*	3	18	6	0	Fr, S, H, U
*Aristichthys nobilis*	2	27	0	0	Fr, S, Fi, U
*Culter oxycephalus*	4	0	0	0	Fr, S, O, U
*Xenocypris davidi*	3	1	0	0	Fr, R, O, L
*Elopichthys bambusa*	1	13	0	0	Fr, S, C, D
*Carassius auratus*	0	5	0	0	Fr, S, O, D
*Ctenopharyngodon idella*	1	8	0	0	Fr, S, H, D
*Hemibarbus maculatus*	1	1	0	0	Fr, S, C, L
*Pseudobrama simoni*	1	0	0	0	Fr, S, O, L
*Myxocyprinus asiaticus*	1	0	0	0	Fr, R, C, L
*Xenocyprisargentea*	3	3	0	0	Fr, S, O, L
*Cultrichthys erythropterus*	0	1	0	0	Fr, S, C, U
*Culter mongolicus*	0	5	0	0	Fr, S, C, L
*Mylopharyngodon piceus*	0	1	0	0	Fr, S, O, L
Perciformes					
*Lateolabrax japonicus*	25	108	80	23	Es, R, C, U
*Odontamblyopus lacepedii*	2	4	3	0	Es, S, O, D
*Channa argus*	1	0	0	0	Fr, S, C, D
*Chaemrichthys stigmatias*	0	1	0	0	Es, S, C, D
*Tridentiger trigonocephalus*	0	1	3	0	Es, S, C, D
*Eleutheronema tetradactylum*	0	2	3	0	Es, S, C, D
*Caranx sexfasciatus*	0	1	0	0	Ma, S, C, U
*Collichthys lucidus*	0	0	78	5	Ma, S, C, D
*Miichthys miiuy*	0	0	7	0	Ma, S, C, D
*Acanthogobius ommaturus*	0	0	15	0	Es, S, C, D
*Boleophthalmus pectinirostris*	0	0	1	0	Es, S, Fi, D
Siluriformes					
*Pelteobaggrus nitidus*	34	12	0	0	Fr, S, C, D
*Leiocassis longirostris*	7	34	0	0	Fr, S, C, L
*Silurus asotus*	22	38	8	2	Fr, S, C, D
Tetraodontiformes					
*Takifugu xanthopterus*	2	14	0	0	Es, S, C, D
*Takifugu obscurus*	0	14	9	1	Mi, S, C, D
Herring order					
*Coilia nasus*	344	475	66	3	Mi, R, C, U
*Coilia mystus*	78	26	0	0	Mi, R, C, U
Pleuroniformes					
*Cynoglossus gracilis*	82	224	54	8	Es, S, Fi, D
*Cynoglossus purpureomaculatus*	1	0	0	0	Ma, S, Fi, D
Mugiliformes					
*Liza haematocheila*	8	177	18	8	Es, S, C, L
*Mugil cephalus*	12	33	44	8	Es, R, O, D
Cucurbitaceae					
*Salanx ariakensis kishinouye*	3	10	0	0	Fr, S, Fi, U
Anguilliformes					
*Anguilla japonica*	2	5	1	0	Mi, R, O, D
Sturgeons					
*hybrid sturgeon*	0	0	1	0	Fr, S, O, L
*Acipenser sinensis*	1	0	0	0	Mi, S, C, L

Note: Mi—migration; Ma—ocean; R—migratory; S—sedentary; O—omnivore; C—carnivore; Fi—filter-feeding; H—herbivore; U—upper; L—lower; D—demersal; Fr—freshwater; Es—estuary.

**Table 2 biology-14-01785-t002:** Seasonal index of relative importance (*IRI*) and dominance rank of dominant Fish species in the southern branch of the Yangtze River Estuary.

Species Name		*IRI*				Dominance Rank (by *IRI*)
		Spring	Summer	Autumn	Winter	
Summer	*Coilia nasus*	6339.41	2999.38		595.66	Spr. Sum. Aut.
	*Cynoglossus gracilis*	1477.01	1672.79		1602.60	Spr. Sum. Aut. Win.
	*Liza haematocheila*	722.10	3005.06		7666.47	Sum. Win.
	*Parabramis pekinensis*	2519.28	2630.96		—	Spr. Sum. Aut.
*Coilia mystus*		1044.24	139.98	—	—	Spr.
*Culter alburnus*		924.78	2290.19	914.46	—	Sum.
*Lateolabrax japonicus*		605.59	1576.38	2163.89	4855.04	Sum. Aut. Win.
*Mugil cephalus*		927.70	769.68	3273.45	2809.52	Aut. Win.
*Pelteobaggrus nitidus*		524.85	68.00	—	—	
*Saurogobio dumerili*		65.88	249.83	107.96	—	
*Leiocassis longirostris*		383.25	568.93	—	—	
*Pseudobrama simoni*		381.62	33.77	47.65	—	
*Cyprinus carpio*		1452.87	1462.72	—	—	Spr. Sum
*Hemiculter bleekeri*		276.79	17.98	—	—	
*Odontamblyopus lacepedii*		25.59	20.74	64.48	—	
*Squaliobarbus curriculus*		165.03	61.83	—	—	
*Anguilla japonica*		72.75	79.67	51.42	—	
*Salanx ariakensis kishinouye*		36.35	51.64	—	—	
*Hypophthalmichthys molitrix*		800.20	257.19	866.43	—	
*Aristichthys nobilis*		204.02	615.77	—	—	
*Culter oxycephalus*		81.06	—	—	—	
*Xenocypris davidi*		63.66	5.18	—	—	
*Elopichthys bambusa*		18.69	237.43	—	—	
*Takifugu xanthopterus*		30.66	83.42	—	—	
*Silurus asotus*		354.45	444.28	856.63	1354.02	Win.
*Carassius auratus*		—	42.52	—	—	
*Ctenopharyngodon idella*		41.34	184.54	—	—	
*Cynoglossus purpureomaculatus*		14.42	—	—	—	
*Channa argus*		93.92	—	—	—	
*Hemibarbus maculatus*		56.00	13.36	—	—	
*Pseudobrama macrops*		60.21	—	—	—	
*Myxocyprinus asiaticus*		17.27	—	—	—	
*Xenocyprisargentea*		49.90	19.80	—	—	
*Cultrichthys erythropterus*		—	6.24	—	—	
*Chaemrichthys stigmatias*		—	5.39	—	—	
*Culter mongolicus*		—	38.16	—	—	
*Mylopharyngodon piceus*		—	7.66	—	—	
*Tridentiger trigonocephalus*		—	5.44	64.20	—	
*Takifugu obscurus*		—	105.55	201.87	195.95	
*Eleutheronema tetradactylum*		—	12.37	91.81	—	
*Caranx sexfasciatus*		—	6.97	—	—	
hybrid sturgeon		—	—	731.36	—	
*Acipenser sinensis*		180.48	—	—	—	
*Collichthys lucidus*		—	—	1862.76	920.72	Aut.
*Miichthys miiuy*		—	—	490.56	—	
*Acanthogobius ommaturus*		—	—	327.48	—	
*Boleophthalmus pectinirostris*		—	—	20.52	—	

Note: “—” indicates that the species was not found in the current season. Dominance rank indicates the seasons in which each species was classified as dominant based on *IRI*.

**Table 3 biology-14-01785-t003:** Fish similarity index of the southern branch of the Yangtze River Estuary in different seasons.

Season	Spring	Summer	Autumn	Winter
Spring	100.00%			
Summer	50.00%	100.00%		
Autumn	31.71%	37.50%	100.00%	
Winter	17.14%	18.92%	38.10%	100.00%

## Data Availability

The original contributions presented in this study are included in the article. Further inquiries can be directed to the corresponding author(s).

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
