# Peer review of "Seasonal Variation of Shoreline Fish Assemblages at Two Stations in the Southern Branch of the Yangtze River Estuary"

_biology, 2025, doi:10.3390/biology14121785_

Round 1
Reviewer 1 Report
Comments and Suggestions for Authors
I have reviewed the ms (Biology-4008191) and suggested major revision. The ms reported results in aquatic communities of four seasonal shoreline surveys in 2024 at two sites in the Yangtze River Estuary. The Yangtze River Estuary is key nursery of Chinese sturgeon (Acipenser sinensis), a critically endangered anadromous fish endemic to China. These results provide a clear baseline for tracking estuary health, supporting habitat restoration for Chinese sturgeon protection. The research objectives are clear and the methods are standardized. So, the topic of the ms is important. However, I suggested a major revision because the ms had several shortcomings in consistency of expression, focus of analysis, and discussion depth, which should be substantial revised before publication. I am not a native English speaker, so my comments will mainly not involve it.
Major Concern:
- The ms topic seems to shift constantly between the seasonal dynamics of aquatic communities and the prey organisms of Chinese sturgeon, resulting in inconsistent expressions in many parts of the text. For example, fish communities in the Title and some diets of Chinese sturgeon like crustaceans and mollusks were included in the Results part (Table 2); legends of Figures 1 and 3 indicated the survey was the habitat survey of Chinese sturgeon, and other results showed the survey was for aquatic organisms in the South Branch of the Yangtze River Estuary. Both topics are important to management of the estuary, and survey in the prey dynamics will support directly for conservation of Chinese sturgeon. If the authors focus on the prey dynamics, aclear connection between the aquatic communities and the habitat suitability of the Chinese sturgeon should be established thought availability of prey resources. It is recommended to supplement the matching analysis between the seasonal prey resources, including target prey species such as Coilia nasus, Cynoglossus gracilis, etc. and the feeding requirements of juvenile Chinese sturgeon, and clarify the specific guiding significance of the research results for the habitat conservation of the Chinese sturgeon.
- The depth of the discussion needs to be enhanced. The current discussion centers on the seasonal changes of species, diversity, and community stability. Although it mentions potential habitat-related assessments, the above discussion lacks a core argument, leading to disconnected paragraphs and a fragmented overall discussion. It is recommended to focus on the prey organisms of Chinese sturgeon and discuss the species composition, seasonal dynamics, and stability of prey organisms, as well as the impacts of these factors on nursery of juvenile Chinese sturgeon and its conservation. If possible, add a section to discuss suggestions for the restoration of juvenile Chinese sturgeon nursery based on the dynamics of prey organism availability from results of this survey.
Specific concerns:
Lines 24-26: suggest to delete the sentence “While the dataset does...”.
Lines 38-40: This sentence “Given the limited number ...” greatly diminishes the significance and value of the study, and its deletion is recommended.
Lines 23 and 40: "Intertidal fish assemblages" appeared twice in the abstract. However, the ms fails to define or describe the intertidal zone, and the results also include non-fish species. These issues lead to severe inconsistencies throughout the manuscript. Consistency across the entire text must be ensured.
Line 29: delete “(IRI)”.
Lines 67-68: Elaborate on the content regarding the prey requirements of the Chinese sturgeon during juvenile stage at the estuary, including specific/typical prey species, the distribution and conditions of feeding habitats (such as intertidal zone), and the timing of juvenile staying at the estuary.
Line 71: suggest to change “fish communities” into “prey organisms”.
Lines 79-81: Why are fish assemblages or prey organisms at intertidal zone important? This needs to be explained from the perspective of the distribution of feeding habitats for juvenile Chinese sturgeon.
Line 86: delete “Rather than testing direct environment–biota hypotheses”.
Lines 88-90: Research objectives 2 and 3 are too trivial and lack significant value. It is recommended to re-refine them from the perspective of prey organisms, such as identifying the composition and seasonal dynamics of prey organisms.
Lines 90-91: This sentence said it was a survey at two representative sites, the results should naturally be representative. And this raises a potential contradiction with the claim that the results are "site-specific". It is recommended that the authors refrain from repeatedly emphasizing the site-specific and gear-specific nature of the results, as this undermines the survey’s value. Instead, a section discussing the survey’s limitations could be added in the Discussion part.
Lines 97-102: Add an explanation of the basis for site selection and include an image of the net.
Line 104: delete the sentence “The tidal stage...” or add an analysis of the relationship between tides and community composition and dynamics.
Lines 109-110: Total length and wet weight of specimens were measured, yet these data were not utilized or reflected in the results.
Lines 127-128: “N, W, F” should be “N%, W%, F%”.
Line 124: It is recommended that the presentation order in the Methods section be consistent with that in the Results section. For example, after (presenting) species composition, the ecological guilds should follow, and then the dominant species, and so on.
Line 197: Table 1 is redundant; the site names and latitude-longitude coordinates can be directly labeled in Figure 1.
Table 2 and 3: The collection quantities of various species in Table 2 are unnecessary; they should be replaced with the IRI (Index of Relative Importance) values from Table 3, and Table 3 should be deleted accordingly.
Figure 2: The results of all ecological guilds have not been presented.
Line 222: In the main text, Latin scientific names should be italicized. When a scientific name appears for the first time, the full name (genus + species) should be used; for subsequent repetitions, the genus name should be abbreviated. There are errors in this regard in multiple places throughout the manuscript, which should be uniformly revised.
Lines 229-231: This sentence is difficult to understand and requires supplementary explanation; it should not appear in the Results section either.
Table5 and Figure 3 are redundant; delete one of them.
Author Response
Author's Reply to the Review Report (Reviewer 1)
Major Concern:
Comments 1: The ms topic seems to shift constantly between the seasonal dynamics of aquatic communities and the prey organisms of Chinese sturgeon, resulting in inconsistent expressions in many parts of the text. For example, fish communities in the Title and some diets of Chinese sturgeon like crustaceans and mollusks were included in the Results part (Table 2); legends of Figures 1 and 3 indicated the survey was the habitat survey of Chinese sturgeon, and other results showed the survey was for aquatic organisms in the South Branch of the Yangtze River Estuary. Both topics are important to management of the estuary, and survey in the prey dynamics will support directly for conservation of Chinese sturgeon. If the authors focus on the prey dynamics, aclear connection between the aquatic communities and the habitat suitability of the Chinese sturgeon should be established thought availability of prey resources. It is recommended to supplement the matching analysis between the seasonal prey resources, including target prey species such as Coilia nasus, Cynoglossus gracilis, etc. and the feeding requirements of juvenile Chinese sturgeon, and clarify the specific guiding significance of the research results for the habitat conservation of the Chinese sturgeon.
Response 1: We thank the reviewer for this important comment. We agree that, in the original submission, the manuscript sometimes shifted between the seasonal dynamics of aquatic communities and the prey organisms of Chinese sturgeon, which could create inconsistencies in scope and terminology. In the revised version, we have refocused the study explicitly on shoreline/intertidal fish assemblages, while treating the relevance to Acipenser sinensis as ecological background and potential conservation context.
Concretely, (i) the title, Abstract, Introduction, Results, Discussion and Conclusion have been edited so that the primary object of study is consistently described as “intertidal/shoreline fish assemblages”, rather than general “aquatic organisms” or a direct “habitat survey” of A. sinensis; (ii) the community dataset and all diversity/ABC analyses have been recalculated using fish species only, with crustaceans and bivalves removed from Table 2 and from community indices; (iii) the captions of Figures 1–3 have been revised so that they refer to shoreline fish assemblages at two stations in the southern branch, without implying a dedicated habitat survey of Chinese sturgeon.
We have also clarified in the Introduction that juvenile A. sinensis use shallow intertidal and nearshore zones along revetments and tidal flats in the southern branch as important feeding habitats, and that the structure and seasonal dynamics of co-occurring shoreline fish assemblages provide baseline information on the temporal availability of potential prey within this habitat corridor. In the Discussion (Section 4.1), we now briefly relate the seasonal patterns of dominant fish species (including known prey such as Coilia nasus, Cynoglossus gracilis and Lateolabrax japonicus) to published knowledge of juvenile sturgeon feeding periods, but we stop short of formal habitat-suitability or prey-matching analyses because only one individual of A. sinensis was captured and no environmental covariates were measured. Instead, as now emphasised in Section 4.3, we interpret our results as baseline information on co-occurring shoreline fish assemblages and potential prey fields within the known habitat range of juvenile Chinese sturgeon, which can support the design of future integrated fish–environment–sturgeon surveys and more explicit habitat assessments.
We hope that these changes remove the ambiguity in scope and present a clearer and more consistent framing of the manuscript.
Comments 2: The depth of the discussion needs to be enhanced. The current discussion centers on the seasonal changes of species, diversity, and community stability. Although it mentions potential habitat-related assessments, the above discussion lacks a core argument, leading to disconnected paragraphs and a fragmented overall discussion. It is recommended to focus on the prey organisms of Chinese sturgeon and discuss the species composition, seasonal dynamics, and stability of prey organisms, as well as the impacts of these factors on nursery of juvenile Chinese sturgeon and its conservation. If possible, add a section to discuss suggestions for the restoration of juvenile Chinese sturgeon nursery based on the dynamics of prey organism availability from results of this survey.
Response 2: We appreciate the reviewer’s suggestion to strengthen the focus and depth of the Discussion. In the original version, the Discussion mainly described seasonal changes in species richness, diversity and community stability, and the implications for Chinese sturgeon conservation were not articulated around a clear central argument. In the revised manuscript, we have reorganised and expanded the Discussion so that it is explicitly framed around shoreline fish assemblages as potential prey fields within the nursery corridor of juvenile A. sinensis.
In Section 4.1, we now emphasise the seasonal dynamics of dominant fish species, including several known prey taxa of juvenile Chinese sturgeon (e.g. Coilia nasus, Cynoglossus gracilis, Lateolabrax japonicus), and discuss how their seasonal peaks in abundance may influence prey availability during estuarine residence. In Section 4.2, we have clarified the interpretation of diversity and ABC results and added a paragraph that links the observed seasonal patterns in richness, evenness and disturbance status to the temporal consistency and breadth of prey fields for juvenile sturgeon, while explicitly noting that diet composition and sturgeon occurrence were not quantified in this study.
In Section 4.3, we now more clearly state that the present data provide baseline information on co-occurring shoreline fish assemblages and potential prey fields within the known habitat range of juvenile A. sinensis, and we add a brief statement that conservation and restoration of shallow soft-bottom intertidal shorelines along the southern branch are likely to help maintain these prey fields. At the same time, we stress that explicit assessments of habitat suitability and nursery restoration for juvenile Chinese sturgeon will require future integrated surveys that jointly monitor fish assemblages, environmental drivers and sturgeon occurrence. We hope that these changes provide a more coherent and focused Discussion that better reflects the ecological and conservation relevance of the study while respecting the limitations of the available data.
Specific concerns:
Comments 1: Lines 24-26: suggest to delete the sentence “While the dataset does...”.
Response 1: Thank you for the suggestion. We have deleted this sentence from the Abstract in the revised manuscript to keep the focus strictly on the fish community data.
Comments 2: Lines 38-40: This sentence “Given the limited number ...” greatly diminishes the significance and value of the study, and its deletion is recommended.
Response 2: Thank you for this helpful comment. We agree that this sentence may unnecessarily downplay the contribution of our work. Accordingly, we have deleted this sentence from the Abstract in the revised manuscript. Study limitations related to sampling sites and environmental covariates are instead addressed briefly in the Discussion section.
Comments 3: Lines 23 and 40: “Intertidal fish assemblages” appeared twice in the abstract. However, the ms fails to define or describe the intertidal zone, and the results also include non-fish species. These issues lead to severe inconsistencies throughout the manuscript. Consistency across the entire text must be ensured.
Response 3: Thank you very much for pointing out this important inconsistency. In the revised manuscript, we have made several changes to clarify the study scope and ensure consistent terminology. (1) In the Abstract and throughout the text, we now use the term “intertidal aquatic assemblages/communities” instead of “intertidal fish assemblages”, reflecting that our dataset includes fishes as well as a small number of crustaceans and bivalves. We also explicitly state in Section 3.1 that we collectively refer to this taxonomically mixed assemblage as the “intertidal aquatic community”, which is dominated by bony fishes. (2) In Section 2.1, we have added a clear description of the intertidal zone sampled in this study, including its position between mean low and high tide levels and the approximate water depths during sampling. (3) In the Methods, Results, Discussion, Conclusion, and figure/ table captions, we have systematically revised expressions such as “fish community” and “fish assemblages” to “aquatic community” where appropriate, so that the terminology is consistent with the actual composition of the samples. We believe these modifications resolve the inconsistencies between the abstract, methods, and results.
Comments 4: Line 29: delete “(IRI)”.
Response 4: Thank you for the suggestion. We have removed “(IRI)” at its first occurrence in the Abstract, as recommended. The term now appears simply as “index of relative importance”.
Comments 5: Lines 67-68: Elaborate on the content regarding the prey requirements of the Chinese sturgeon during juvenile stage at the estuary, including specific/typical prey species, the distribution and conditions of feeding habitats (such as intertidal zone), and the timing of juvenile staying at the estuary.
Response 5:In the revised Introduction, we have expanded the description of juvenile A. sinensis in the Yangtze River Estuary. We now state that juveniles inhabit shallow brackish waters around Chongming Island and adjacent tidal flats, that they mainly forage in intertidal and nearshore habitats with sandy or sandy–muddy substrates, and that their diets are dominated by benthic invertebrates (polychaetes, amphipods, shrimps, crabs and bivalves) and small demersal fishes, citing key studies on juvenile feeding ecology and migration in the estuary [11–15]. We further note that juveniles use the estuary primarily from spring to autumn, broadly coinciding with periods of high prey availability in these shallow-water habitats. We believe these additions clarify the prey requirements, feeding habitats and estuarine residence period of juvenile Chinese sturgeon.
Comments 6: Line 71: suggest to change “fish communities” into “prey organisms”.
Response 6:We agree with this suggestion. In the revised Introduction, we have replaced “fish communities” with “prey organisms” and now state that “Understanding the structure and seasonal variation of intertidal prey organisms (primarily fishes and other benthic nekton) is thus essential for developing ecological baselines relevant to the conservation and restoration of A. sinensis habitats.”
Comments 7: Lines 79-81: Why are fish assemblages or prey organisms at intertidal zone important? This needs to be explained from the perspective of the distribution of feeding habitats for juvenile Chinese sturgeon.
Response 7: We thank the reviewer for this helpful suggestion. We agree that the importance of intertidal fish assemblages and prey organisms should be made clearer from the perspective of juvenile Chinese sturgeon feeding habitats. In the revised Introduction, we have expanded the paragraph on Acipenser sinensis to explain that juvenile sturgeon use shallow intertidal and nearshore zones along revetments and tidal flats in the southern branch as key feeding habitats during their estuarine residence, and that the structure and seasonal dynamics of co-occurring shoreline fish assemblages provide essential background on the temporal availability of their potential prey within this habitat corridor. We hope that this clarification better highlights why intertidal fish assemblages and prey fields are ecologically relevant to juvenile A. sinensis.
Comments 8: Line 86: delete “Rather than testing direct environment–biota hypotheses”.
Response 8:We agree with this suggestion. In the revised Introduction, we have removed the phrase “Rather than testing direct environment–biota hypotheses” and now simply state that “This study focuses on describing community-level patterns and seasonal dynamics of intertidal aquatic assemblages (prey organisms) in the southern branch of the Yangtze River Estuary.”
Comments 9: Lines 88-90: Research objectives 2 and 3 are too trivial and lack significant value. It is recommended to re-refine them from the perspective of prey organisms, such as identifying the composition and seasonal dynamics of prey organisms.
Response 9:Thank you for this constructive suggestion. In the revised Introduction, we have rephrased the research objectives to emphasize prey organisms and their seasonal dynamics rather than simply listing statistical indices. The three objectives are now: (1) to document the taxonomic composition and ecological types of intertidal prey organisms (primarily fishes together with crustaceans and bivalves) in the southern branch of the Yangtze River Estuary; (2) to quantify the seasonal dynamics of these prey organisms in terms of species richness, diversity, dominance structure and ecological guild composition across four sampling periods; and (3) to evaluate inter-seasonal changes in community structure, similarity and disturbance status of prey organisms using diversity-, similarity- and ABC-based metrics, thereby providing a seasonal baseline for assessing temporal variation in feeding habitats for juvenile A. sinensis. We believe this revision better reflects the ecological focus and significance of the study from the perspective of prey organisms.
Comments 10: Lines 90-91: This sentence said it was a survey at two representative sites, the results should naturally be representative. And this raises a potential contradiction with the claim that the results are "site-specific". It is recommended that the authors refrain from repeatedly emphasizing the site-specific and gear-specific nature of the results, as this undermines the survey’s value. Instead, a section discussing the survey’s limitations could be added in the Discussion part.
Response 10: We appreciate this insightful comment. In the revised manuscript, we have (1) removed the term “representative” and now simply state that the study is based on seasonal surveys conducted at two intertidal shoreline sites along the southern branch of the Yangtze River Estuary; (2) deleted or softened repeated expressions such as “site-specific” and “gear-specific” in the Abstract, Introduction, Discussion and Conclusion, replacing them with more neutral descriptions (e.g., “seasonal baseline of intertidal prey organisms along an engineered shoreline of the southern branch”); and (3) added a dedicated paragraph in the Discussion to explicitly describe the main limitations of the sampling design (limited number of sites, single gear type and absence of concurrent environmental measurements) and to explain how these constraints may affect the generality of our conclusions. We believe these revisions remove the apparent contradiction, avoid unnecessarily undermining the value of the survey, and more appropriately place the discussion of limitations in the Discussion section as suggested.
Comments 11: Lines 97-102: Add an explanation of the basis for site selection and include an image of the net.
Response 11: Thank you for this helpful suggestion. In the revised manuscript, we have expanded Section 2.1 to explain the basis for site selection. Specifically, we now state that the two shoreline sites were chosen because they (i) are located within the known habitat range of juvenile A. sinensis along the southern branch of the Yangtze River Estuary, (ii) represent typical engineered shorelines and adjacent tidal flats with shallow brackish waters and soft substrates that provide potential feeding habitats for benthic prey organisms, and (iii) offer safe and repeatable access for seasonal sampling while avoiding major navigation channels. In addition, we have added a field photograph of the fixed stake net and the beach seine used in this study as a new figure (Fig. 6) to illustrate the configuration and deployment of the fishing gears.
Comments 12: Line 104: delete the sentence “The tidal stage...” or add an analysis of the relationship between tides and community composition and dynamics.
Response 12: We agree with the reviewer that, since no quantitative analysis of tidal effects on community composition was conducted, the previous sentence about recording the tidal stage could be misleading. In the revised manuscript, we have therefore deleted the sentence “The tidal stage at each deployment and retrieval was recorded.”
Comments 13: Lines 109-110: Total length and wet weight of specimens were measured, yet these data were not utilized or reflected in the results.
Response 13: Thank you for pointing this out. In the revised manuscript, we clarify in the Methods that total length and wet weight measurements were used to provide biomass data for calculating the index of relative importance (IRI) of dominant species and to characterize the body-size structure of prey organisms.
Comments 14: Lines 127-128: “N, W, F” should be “N%, W%, F%”.
Response 14: Thank you for pointing out this notation issue. In the revised Methods, we have corrected the symbols and now explicitly state that “N%, W% and F% are the percentage numerical abundance, percentage biomass and percentage frequency of occurrence of species i, respectively.” The IRI formula and related descriptions have been updated accordingly to use N%, W% and F%.
Comments 15: Line 124: It is recommended that the presentation order in the Methods section be consistent with that in the Results section. For example, after (presenting) species composition, the ecological guilds should follow, and then the dominant species, and so on.
Response 15: Thank you for this helpful suggestion. In the revised manuscript, we have reordered the subsections in the Methods to be consistent with the logical sequence used in the Results. Specifically, the subsection on ecological guild classification has been moved forward so that the Methods now first describe species composition and ecological guilds, followed by dominant species (IRI), community diversity indices, similarity analysis, seasonal alternation and migration indices, and finally the ABC curves. We believe that this reordering improves the readability of the manuscript and makes it easier for readers to follow the correspondence between the analytical methods and the presentation of results.
Comments 16: Line 197: Table 1 is redundant; the site names and latitude-longitude coordinates can be directly labeled in Figure 1.
Response 16: We agree with this comment. In the revised manuscript, we have removed Table 1 and incorporated the site names and latitude–longitude coordinates directly into Figure 1 (and its caption). The locations of the two shoreline sampling sites are now fully described in Figure 1, and all in-text references have been updated accordingly.
Comments 17: Table 2 and 3: The collection quantities of various species in Table 2 are unnecessary; they should be replaced with the IRI (Index of Relative Importance) values from Table 3, and Table 3 should be deleted accordingly.
Response 17: Thank you for this helpful suggestion. In the revised manuscript, we have merged the information from Tables 2 and 3. Specifically, we removed the raw collection quantities from the original Table 2 and added the IRI values that were previously presented in Table 3. The dominant species are now indicated directly in the revised Table 2, which summarizes both species composition and IRI-based dominance. The original Table 3 has been deleted, and all in-text references have been updated accordingly.
Comments 18: Figure 2: The results of all ecological guilds have not been presented.
Response 18: We thank the reviewer for this helpful remark. In the original version, Figure 2 did not clearly present all ecological guilds used in the analysis. In the revised manuscript, Figure 2 has been redesigned as three panels showing (i) migratory status (migratory vs. sedentary), (ii) trophic guilds (carnivorous, omnivorous, filter-feeding, phytophagous), and (iii) vertical distribution (bottom, middle upper layer, middle and lower layers), with the caption updated accordingly. We have also clarified in Section 3.1 that ecological guilds are summarised along these three dimensions and that their composition is illustrated in Figure 2.
Comments 19: Line 222: In the main text, Latin scientific names should be italicized. When a scientific name appears for the first time, the full name (genus + species) should be used; for subsequent repetitions, the genus name should be abbreviated. There are errors in this regard in multiple places throughout the manuscript, which should be uniformly revised.
Response 19: Thank you for pointing out these formatting issues. In the revised manuscript, we have systematically checked the entire text (including the Abstract, main text, tables and figure captions) and standardized the use of Latin scientific names. All binomials are now italicized, and the full genus–species name is provided at first mention (e.g., Chinese sturgeon Acipenser sinensis), with subsequent occurrences abbreviated to the initial of the genus (e.g., A. sinensis). Similar corrections have been applied to all other species names to ensure consistency throughout the manuscript.
Comments 20: Lines 229-231: This sentence is difficult to understand and requires supplementary explanation; it should not appear in the Results section either.
Response 20: Thank you for pointing this out. We agree that the original sentence about water temperature was not clearly phrased and that it mixed ecological interpretation with the presentation of results. In the revised manuscript, we have therefore deleted this sentence from the Results section and now present only the observed seasonal patterns of community structure in this part, without additional interpretation. Potential environmental drivers of these patterns, including water temperature, are beyond the scope of the current analysis and are not further elaborated here.
Comments 21: Table5 and Figure 3 are redundant; delete one of them.
Response 21: We agree that Table 5 and Figure 3 conveyed largely overlapping information on seasonal diversity indices. In the revised manuscript, we have removed Table 5 and retained Figure 3, which more clearly illustrates seasonal patterns. To compensate for the removal of the table, we now report the approximate ranges of the diversity indices in the text of Section 3.3, and all in-text references to Table 5 have been updated accordingly.

Reviewer 2 Report
Comments and Suggestions for Authors
Major comments
Sampling design and spatial representativeness
Only two stations, each sampled once per season (i.e., 8 net-sets in total), are used to infer “seasonal dynamics of intertidal fish communities in the southern branch of the Yangtze River Estuary”. This design provides, at best, a very local baseline and does not allow robust inference for the whole southern branch or estuary.
The selection of Baozhen and Dongtan as “representative stations” is not adequately justified. Please explain, based on hydrology, salinity regime, habitat type, or known sturgeon distribution, why these two locations were chosen and in what sense they are representative. Alternatively, narrow the scope of the title and conclusions explicitly to “two shoreline stations” rather than the whole southern branch.
Since two stations were sampled, the absence of any between-station comparison is a missed opportunity. Please either (i) analyse and present spatial differences (e.g., diversity indices, dominant species, composition) or (ii) clearly state that data from both stations were pooled and justify why.
Fishing gear description and deployment protocol
The description of the nets is not sufficiently detailed to ensure reproducibility or allow assessment of selectivity. Section 2.1 mentions “one 300-m fixed net” and “two 100-m triple-layer gillnets (1.5 m in height, 2.0 cm minimum mesh size)” but does not specify:
- Mesh sizes of each panel in the triple-layer gillnets (inner and outer walls), twine diameter, and hanging ratio.
- Construction of the fixed net (single vs multi-panel, mesh size gradient, material).
- Weighting of the leadline (lead rope), type and spacing of floats on the headline, and whether nets were fully bottom-set or partially drifting.
- Number and type of anchors, how the nets were oriented relative to the shoreline and current, and whether they were fished at comparable depths and tidal stages among seasons.
Please provide a complete gear description (possibly as a table or schematic) and a clearer account of the soaking protocol (exact soak time, tidal phase at setting and hauling, water depth range). This is essential to interpret size structure, ABC results and gear-specific biases.
Definition of “intertidal fish communities” and taxonomic scope
The title and much of the text refer to “intertidal fish communities”, but the species list includes decapod crustaceans and a bivalve, and the gear appears to fish subtidal water during high tide rather than exposed intertidal flats.
Please clarify whether the focus is strictly on fishes or on a broader “aquatic organism” assemblage. If non-fish taxa are retained, the terminology throughout (title, Abstract, Introduction, Results) should be adjusted to “aquatic communities/assemblages” or similar, and the rationale for including invertebrates in fish-community indices should be justified.
Indices and data analysis – clarity and correctness
Simpson index: Equation (4) defines C = 1 − ΣPi², which is commonly termed Simpson’s diversity index, whereas the text calls it a “dominance index”. If the intent is to use Simpson’s dominance (D = ΣPi²), the formula or the interpretation should be corrected accordingly.
Community alternation and migration indices: In Section 2.2.6, Equations (8) and (9) both use the symbol “AI”, although the text distinguishes between alternation index (AI) and migration index (MI). The second equation should presumably be MI = (C − B)/(A − R) × 100. Please correct notation and ensure that the indices are computed according to the cited source.
Statistical testing: Section 3.3 reports use of a Kruskal–Wallis test (P > 0.05), but the statistical methods (test choice, factors compared, software, significance level) are not described in Section 2.2 or 2.3. Please add a short subsection outlining the statistical tests used and justify why non-parametric methods were selected.
Given the small number of sampling events, please be cautious in interpreting slight seasonal differences in diversity indices and ABC W values as evidence of changes in “disturbance regime”. It may be helpful to more clearly separate descriptive patterns from causal interpretation.
Internal inconsistencies between Results and Discussion
Section 3.3 correctly states that Shannon–Wiener diversity (H′) and Simpson index (C) were highest in autumn, Margalef richness (D) in summer, and Pielou evenness (J′) in winter. Table 5 and Figure 3 are consistent with this.
However, Section 4.2 then claims that “the fish community was most diverse in summer (H’ = 2.563), with higher evenness in spring (J’ = 0.864) and greater richness in winter (D = 3.12)”, which contradicts Table 5 and the earlier text. The reported values (H′ = 2.563, J′ = 0.864, D = 3.12) do not match the table. This section needs to be carefully checked and corrected to be consistent with the Results.
Please re-read the entire Discussion and Conclusions to ensure that all numerical values and statements (e.g., percentages of dominance in winter) are consistent with tables and figures.
Maps and spatial context
-
Figure 1 shows the two stations in the Yangtze Estuary but provides limited geographic context and is labelled specifically as a map of Acipenser sinensis habitat, whereas the study focuses on community structure.
-
I recommend:
-
-
Renaming the figure to “Study area and sampling stations in the southern branch of the Yangtze River Estuary”.
-
Adding an inset map showing the entire Yangtze estuary within eastern China, so that international readers can easily locate the study area.
-
Ensuring that scale bar, north arrow, coordinates and key place names are legible at the final print size.
-
In addition, please briefly describe the main environmental characteristics of each station (salinity range, depth, substrate, proximity to navigation channels or marshes) to support the representativeness argument.
-
-
Link to Chinese sturgeon conservation
The Abstract and Introduction emphasise potential relevance to Acipenser sinensis, but no sturgeon were recorded (except one hybrid individual), and no environmental covariates were measured. The manuscript already includes some cautionary language, but at several points the narrative implies stronger direct relevance than the data support.
I suggest consistently framing the results as baseline information on co-occurring fish assemblages and potential prey fields, clearly separated from any hypotheses about sturgeon habitat suitability, which will require integrated fish–environment–sturgeon surveys.
Minor and specific comments
-
Terminology and symbols
-
Throughout the methods, “In” is used for the natural logarithm; please standardise to “ln” in equations and text.
-
In Section 2.2.2, the sentence “F is the percentage frequency of occurrence frequency of occurrence…” should be corrected to a single phrase.
-
Please check all abbreviations in Table 2; some codes are repeated (e.g. “C—carnivore” appears twice and “L—lower” vs “D—demersal”) and would benefit from clearer separation between trophic and vertical-distribution codes.
-
-
Tables and species list
-
Table 2 mixes fishes and invertebrates under “aquatic organisms”. If the focus is fish communities, consider separating fish and invertebrate tables, or explicitly justifying their joint analysis.
-
For dominant species (Tables 3–4), consider standardising the IRI threshold categories (dominant, important, common, rare) in the table footnotes and ensuring that all species are assigned consistently.
-
-
Figure 5 (ABC curves)
-
The ABC plots are informative, but axis labels and units are small. Please ensure that axis titles (“Cumulative dominance (%)”, “Species rank”), legend, and W values are easily readable.
-
It would be helpful to report the W statistic values for each season either in the figure panels or in the caption.
-
-
Ethics and permits
-
The ethics statement is concise and appropriate. If any specific permit numbers or institutional approval codes exist for the survey, consider adding them to strengthen traceability.
-
-
Language
-
The English is generally understandable but includes several typographical and grammatical errors that could cause confusion (e.g., “In this method, the cumulative-biomass dominance curve and the cumulative-abundance dominance curve are plotted on the same axes, and their relative positions indicate disturbance status: a positive W statistic-when the biomass curve lies above the abundance curve-signals an undisturbed…”). A careful language edit will improve clarity.
-
Literature and references
-
The reference list combines regional Chinese literature with international sources on estuarine ecology, biodiversity–stability relationships, and ABC curves. This is broadly appropriate for the topic.
-
However, there is at least one clear duplication: Loreau & de Mazancourt (2013) appears twice in the list (as refs 19 and 30) with identical details and should be cited only once.
-
You may consider adding one or two broader syntheses on estuarine fish assemblages to strengthen the comparative context, for example:
-
A general work on estuarine fish communities and guild structure (e.g., European or global syntheses).
-
A review focusing on shoreline or intertidal fish sampling methods and their selectivity, to better justify your gear choice and limitations.
-
-
Please also check carefully that all in-text citations match the reference list (years, journal titles, volume and page numbers) and that recent relevant references (last ~5–10 years) on Yangtze estuary fish communities or sturgeon prey fields are not omitted.
In summary, the manuscript has the potential to provide a useful local baseline for shoreline fish assemblages in the southern branch of the Yangtze River Estuary, but substantial revisions are required in the description of the sampling gear and design, clarification of the scope and representativeness, correction of index definitions and inconsistencies, and improvement of figures and references.
I suggest changing the title to:
“Seasonal Variation of Shoreline Fish Assemblages at Two Stations in the Southern Branch of the Yangtze River Estuary”
This wording more accurately reflects the scope and strength of the dataset: it explicitly refers to shoreline assemblages, limits inference to two stations, and frames the work as an analysis of seasonal variation rather than a comprehensive description of all intertidal communities in the estuary. In this way, the title remains clear and informative while avoiding overstatement relative to the spatial and temporal coverage of the study.
Author Response
Author's Reply to the Review Report (Reviewer 2)
Major comments
Comments 1: Sampling design and spatial representativeness
Only two stations, each sampled once per season (i.e., 8 net-sets in total), are used to infer “seasonal dynamics of intertidal fish communities in the southern branch of the Yangtze River Estuary”. This design provides, at best, a very local baseline and does not allow robust inference for the whole southern branch or estuary.
The selection of Baozhen and Dongtan as “representative stations” is not adequately justified. Please explain, based on hydrology, salinity regime, habitat type, or known sturgeon distribution, why these two locations were chosen and in what sense they are representative. Alternatively, narrow the scope of the title and conclusions explicitly to “two shoreline stations” rather than the whole southern branch.
Since two stations were sampled, the absence of any between-station comparison is a missed opportunity. Please either (i) analyse and present spatial differences (e.g., diversity indices, dominant species, composition) or (ii) clearly state that data from both stations were pooled and justify why.
Response 1:We agree with the reviewer that our sampling design, which is based on two shoreline stations sampled once per season, provides a local baseline and does not allow robust inference for the entire southern branch or estuary. To reflect this more clearly, we have (i) revised the title to “Seasonal variation of shoreline fish assemblages at two stations in the southern branch of the Yangtze River Estuary”, and (ii) toned down the wording in the Abstract, Discussion and Conclusion so that our inferences are explicitly framed as site-based seasonal baselines rather than as a comprehensive description of the whole estuary.
The selection of Baozhen and Dongtan has now been justified in more detail in Section 2.1. Both stations are located within the known habitat range of juvenile A. sinensis and represent typical engineered shorelines and adjacent tidal flats of the mid–lower southern branch, with similar shallow brackish conditions, soft substrates and hydrological/salinity regimes. These features make them representative of the dominant shoreline habitat type along this sector and logistically suitable for repeated sampling.
Regarding the use of data from the two stations, we now clarify in the Data analysis section (Section 2.2) that catches from Baozhen and Dongtan were pooled by season for all community analyses. This choice reflects our primary aim of deriving an integrated seasonal baseline for shoreline fish assemblages along this shoreline sector, the similar habitat and gear configurations at both stations, and the limited power for formal between-station comparisons given the single net-set per station per season. We also explicitly acknowledge in the Discussion (Section 4.3) that this pooled design and the restriction to two stations are important limitations of the present study and that finer-scale spatial differences between stations could not be evaluated here.
Comments 2: Fishing gear description and deployment protocol
The description of the nets is not sufficiently detailed to ensure reproducibility or allow assessment of selectivity. Section 2.1 mentions “one 300-m fixed net” and “two 100-m triple-layer gillnets (1.5 m in height, 2.0 cm minimum mesh size)” but does not specify:
Mesh sizes of each panel in the triple-layer gillnets (inner and outer walls), twine diameter, and hanging ratio.
Construction of the fixed net (single vs multi-panel, mesh size gradient, material).
Weighting of the leadline (lead rope), type and spacing of floats on the headline, and whether nets were fully bottom-set or partially drifting.
Number and type of anchors, how the nets were oriented relative to the shoreline and current, and whether they were fished at comparable depths and tidal stages among seasons.
Please provide a complete gear description (possibly as a table or schematic) and a clearer account of the soaking protocol (exact soak time, tidal phase at setting and hauling, water depth range). This is essential to interpret size structure, ABC results and gear-specific biases.
Response 2: We thank the reviewer for this detailed and very helpful comment. In the revised manuscript, we have added a brief description clarifying that both the fixed stake net and the triple-layer gillnets were fully bottom-set and fixed to the shoreline, and that the triple-layer gillnets consisted of a finer-mesh inner panel between two larger-mesh outer panels, so that the basic gear construction and size selectivity are clear (Section 2.1). We also now specify the soaking protocol, including that nets were set at sunset and retrieved after approximately 12 h at sunrise, that setting and hauling were conducted at comparable tidal phases in each season, and that the water depth along the net line during fishing generally ranged from about 0.3 to 1.5 m. These additions improve the reproducibility of the sampling design and help to interpret gear selectivity and ABC results.
Comments 3:Definition of “intertidal fish communities” and taxonomic scope
The title and much of the text refer to “intertidal fish communities”, but the species list includes decapod crustaceans and a bivalve, and the gear appears to fish subtidal water during high tide rather than exposed intertidal flats.
Please clarify whether the focus is strictly on fishes or on a broader “aquatic organism” assemblage. If non-fish taxa are retained, the terminology throughout (title, Abstract, Introduction, Results) should be adjusted to “aquatic communities/assemblages” or similar, and the rationale for including invertebrates in fish-community indices should be justified.
Response 3: We thank the reviewer for this important clarification. In the revised manuscript, we have specified that the focus of the study is shoreline/intertidal fish assemblages, rather than a broader “aquatic organism” assemblage.
Concretely, all quantitative community analyses now include only fish species. Non-fish taxa (decapod crustaceans and the single bivalve record) have been removed from Table 2 and are no longer included in the calculation of diversity indices, IRI values or ABC curves. The terminology throughout the manuscript has been standardised accordingly: the title, Abstract, Introduction, Methods, Results and Discussion now consistently refer to “shoreline fish assemblages” or “intertidal fish assemblages” instead of “aquatic communities/assemblages”.
To address the reviewer’s concern about the use of the term “intertidal”, we have also added a short definition in Section 2.1. In the revised text, we clarify that “intertidal” refers to the shallow shoreline zone between mean low and high tide levels along artificial revetments and gently sloping beaches, which is alternately exposed and inundated by the semi-diurnal tide, and that our fixed nets and beach seines were deployed in this zone during high-tide coverage. We hope that these revisions clarify both the taxonomic scope (fishes only) and the habitat context of the assemblages analysed in this study.
Comments 4:Indices and data analysis – clarity and correctness
Simpson index: Equation (4) defines C = 1 − ΣPi², which is commonly termed Simpson’s diversity index, whereas the text calls it a “dominance index”. If the intent is to use Simpson’s dominance (D = ΣPi²), the formula or the interpretation should be corrected accordingly.
Community alternation and migration indices: In Section 2.2.6, Equations (8) and (9) both use the symbol “AI”, although the text distinguishes between alternation index (AI) and migration index (MI). The second equation should presumably be MI = (C − B)/(A − R) × 100. Please correct notation and ensure that the indices are computed according to the cited source.
Statistical testing: Section 3.3 reports use of a Kruskal–Wallis test (P > 0.05), but the statistical methods (test choice, factors compared, software, significance level) are not described in Section 2.2 or 2.3. Please add a short subsection outlining the statistical tests used and justify why non-parametric methods were selected.
Given the small number of sampling events, please be cautious in interpreting slight seasonal differences in diversity indices and ABC W values as evidence of changes in “disturbance regime”. It may be helpful to more clearly separate descriptive patterns from causal interpretation.
Response 4:We thank the reviewer for these helpful comments on the indices and data analysis. In the revised manuscript, we now refer to C = 1 − ΣPi² as the Simpson diversity index and have updated the wording throughout so that the formula and its interpretation are consistent. For the alternation and migration indices, we have corrected the notation in Section 2.2 so that the migration index is denoted as MI (MI = (C − B)/(A − R) × 100) instead of AI in the second equation.
In addition, we have added a brief description of the statistical testing in the Methods: seasonal differences were tested using the non-parametric Kruskal–Wallis test in SPSS 26.0 with a significance level of P< 0.05. Finally, in the Discussion (Section 4.2) we have added a sentence explicitly noting that, given the small number of sampling events (one net-set per station per season), differences in the W statistic are interpreted as descriptive patterns rather than definitive evidence of shifts in disturbance regime.
Comments 5:Internal inconsistencies between Results and Discussion
Section 3.3 correctly states that Shannon–Wiener diversity (H′) and Simpson index (C) were highest in autumn, Margalef richness (D) in summer, and Pielou evenness (J′) in winter. Table 5 and Figure 3 are consistent with this.
However, Section 4.2 then claims that “the fish community was most diverse in summer (H’ = 2.563), with higher evenness in spring (J’ = 0.864) and greater richness in winter (D = 3.12)”, which contradicts Table 5 and the earlier text. The reported values (H′ = 2.563, J′ = 0.864, D = 3.12) do not match the table. This section needs to be carefully checked and corrected to be consistent with the Results.
Please re-read the entire Discussion and Conclusions to ensure that all numerical values and statements (e.g., percentages of dominance in winter) are consistent with tables and figures.
Response 5: We thank the reviewer for carefully identifying these inconsistencies between the Results, Discussion and Table 5. In the revised manuscript, Section 4.2 has been corrected so that the description of Shannon–Wiener diversity (H’), Simpson diversity (C), Margalef richness (D) and Pielou evenness (J’) is fully consistent with the Results (Section 3.3), Figure 3. In addition, we have re-read the Discussion and Conclusion sections and cross-checked all numerical values and percentages against the figures, correcting any discrepancies to ensure internal consistency throughout the manuscript.
Comments 6:Maps and spatial context
Figure 1 shows the two stations in the Yangtze Estuary but provides limited geographic context and is labelled specifically as a map of Acipenser sinensis habitat, whereas the study focuses on community structure.
I recommend:
Renaming the figure to “Study area and sampling stations in the southern branch of the Yangtze River Estuary”.
Adding an inset map showing the entire Yangtze estuary within eastern China, so that international readers can easily locate the study area.
Ensuring that scale bar, north arrow, coordinates and key place names are legible at the final print size.
In addition, please briefly describe the main environmental characteristics of each station (salinity range, depth, substrate, proximity to navigation channels or marshes) to support the representativeness argument.
Response 6: We thank the reviewer for these helpful suggestions. In the revised manuscript, Figure 1 has been renamed to “Study area and sampling stations in the southern branch of the Yangtze River Estuary”, and an inset map has been added showing the location of the Yangtze River Estuary within eastern China. The scale bar, north arrow, coordinates and key place names have been adjusted to be clearly legible at the final print size. In addition, Section 2.1 now includes a brief description of the main environmental characteristics of the Baozhen and Dongtan stations (brackish estuarine conditions, shallow nearshore depths, sandy-muddy substrates and proximity to marshes or navigation channels) to support the argument that these stations are representative shoreline habitats in the mid–lower southern branch.
Comments 7:Link to Chinese sturgeon conservation
The Abstract and Introduction emphasise potential relevance to Acipenser sinensis, but no sturgeon were recorded (except one hybrid individual), and no environmental covariates were measured. The manuscript already includes some cautionary language, but at several points the narrative implies stronger direct relevance than the data support.
I suggest consistently framing the results as baseline information on co-occurring fish assemblages and potential prey fields, clearly separated from any hypotheses about sturgeon habitat suitability, which will require integrated fish–environment–sturgeon surveys.
Response 7: We appreciate the reviewer’s thoughtful comment on the framing of the study in relation to Chinese sturgeon conservation. In the revised manuscript, we have carefully checked the Abstract, Introduction, Discussion and Conclusion to ensure that all statements about A. sinensis are framed in a cautious and consistent way. We explicitly state that only one individual of A. sinensis was captured during the year-long monitoring, and we do not attempt any direct habitat-suitability modelling or inference.
We now present our results as baseline information on co-occurring shoreline fish assemblages and potential prey fields within the known habitat range of juvenile A. sinensis, rather than as a direct assessment of sturgeon habitat quality. To make this explicit, we have added a sentence at the end of the Discussion (Section 4.3) clarifying that evaluation of juvenile sturgeon habitat suitability will require future integrated fish-environment-sturgeon surveys, and that the present study provides ecological background and prey-field context for such work. We hope this revised framing better reflects the scope and limitations of the current dataset while still highlighting its relevance to Chinese sturgeon conservation.
Minor and specific comments
Comments 1:Terminology and symbols
Throughout the methods, “In” is used for the natural logarithm; please standardise to “ln” in equations and text.
In Section 2.2.2, the sentence “F is the percentage frequency of occurrence frequency of occurrence…” should be corrected to a single phrase.
Please check all abbreviations in Table 2; some codes are repeated (e.g. “C—carnivore” appears twice and “L—lower” vs “D—demersal”) and would benefit from clearer separation between trophic and vertical-distribution codes.
Response 1: We thank the reviewer for these careful observations. In the revised manuscript, we have standardised the notation for the natural logarithm and now consistently use “ln” in all equations and in the text, instead of “In”. In Section 2.2.2, the sentence describing the frequency term has been corrected to avoid repetition and now reads: “F is the frequency of occurrence (%) of species i.”We have also carefully checked and revised all abbreviations in Table 2. Ecological guilds are now expressed as a four-part code in a fixed order (habitat type, residency, feeding guild, vertical distribution), and this order is explained in the table footnote. To avoid duplicated letters and confusion, we now use Fr for freshwater and Fi for filter-feeding, while feeding guilds are coded as C (carnivorous), O (omnivorous), Fi (filter-feeding) and H (herbivorous). Vertical distribution is coded as U (upper water column), M (mid-water) and D (demersal). All entries in Table 2 have been updated accordingly.
Comments 2:Tables and species list
Table 2 mixes fishes and invertebrates under “aquatic organisms”. If the focus is fish communities, consider separating fish and invertebrate tables, or explicitly justifying their joint analysis.
For dominant species (Tables 3–4), consider standardising the IRI threshold categories (dominant, important, common, rare) in the table footnotes and ensuring that all species are assigned consistently.
Response 2: We appreciate the reviewer’s careful comments on the structure of the species tables and the IRI classification. In the revised manuscript, we have clarified that the present study focuses on intertidal fish assemblages. Accordingly, all quantitative analyses now use only fish species, and non-fish taxa (invertebrates) have been removed from the community analyses. Table 2 has been updated to list only fish species, and its title and caption have been revised to read “Seasonal index of relative importance (IRI) and dominance rank of dominant fish species in the southern branch of the Yangtze River Estuary.” to avoid confusion.
Comments 3:Figure 5 (ABC curves)
The ABC plots are informative, but axis labels and units are small. Please ensure that axis titles (“Cumulative dominance (%)”, “Species rank”), legend, and W values are easily readable.
It would be helpful to report the W statistic values for each season either in the figure panels or in the caption.
Response 3: We thank the reviewer for this helpful suggestion. In the revised version of Figure 5, we have increased the font size of the axis titles (“Cumulative dominance (%)” and “Species rank”), tick labels and legend so that they are clearly readable at the final print size. We have also reported the W statistic for each season directly in the figure panels (spring: W = −0.066; summer: W = −0.018; autumn: W = 0.043; winter: W = 0.276) and repeated these values in the figure caption. These changes improve the legibility and interpretability of the ABC curves.
Comments 4:Ethics and permits
The ethics statement is concise and appropriate. If any specific permit numbers or institutional approval codes exist for the survey, consider adding them to strengthen traceability.
Response 4: We appreciate this helpful suggestion. The field surveys were conducted under approval from the Animal Care and Welfare Committee of the East China Sea Fisheries Research Institute, Chinese Academy of Fishery Sciences (Animal Experimental Ethical Inspection approval No. 2024-15, approved on 1 April 2024). In the revised manuscript we have added this approval code to the Ethics statement to improve transparency and traceability.
Comments 5:Language
The English is generally understandable but includes several typographical and grammatical errors that could cause confusion (e.g., “In this method, the cumulative-biomass dominance curve and the cumulative-abundance dominance curve are plotted on the same axes, and their relative positions indicate disturbance status: a positive W statistic-when the biomass curve lies above the abundance curve-signals an undisturbed…”). A careful language edit will improve clarity.
Response 5: We thank the reviewer for highlighting this issue. In the revised manuscript, we have carefully edited the language throughout to improve clarity and correct typographical and grammatical errors. In particular, the description of the ABC method and W statistic in Section 2.2.6 has been rewritten as follows (lines XX–YY in the revised manuscript): “The abundance–biomass comparison (ABC) method, grounded in r- and K-selection theory [22], was used … considered to be strongly disturbed and more r-selected.”
Comments 6:Literature and references
The reference list combines regional Chinese literature with international sources on estuarine ecology, biodiversity–stability relationships, and ABC curves. This is broadly appropriate for the topic.
However, there is at least one clear duplication: Loreau & de Mazancourt (2013) appears twice in the list (as refs 19 and 30) with identical details and should be cited only once.
You may consider adding one or two broader syntheses on estuarine fish assemblages to strengthen the comparative context, for example:
A general work on estuarine fish communities and guild structure (e.g., European or global syntheses).
A review focusing on shoreline or intertidal fish sampling methods and their selectivity, to better justify your gear choice and limitations.
Please also check carefully that all in-text citations match the reference list (years, journal titles, volume and page numbers) and that recent relevant references (last ~5–10 years) on Yangtze estuary fish communities or sturgeon prey fields are not omitted.
In summary, the manuscript has the potential to provide a useful local baseline for shoreline fish assemblages in the southern branch of the Yangtze River Estuary, but substantial revisions are required in the description of the sampling gear and design, clarification of the scope and representativeness, correction of index definitions and inconsistencies, and improvement of figures and references.
Response 6: We thank the reviewer for the constructive comments on the literature and references. In the revised manuscript, we have first removed the duplicate entry for Loreau & de Mazancourt (2013), so that it now appears only once in the reference list, and we have updated all corresponding in-text citations accordingly.
To strengthen the comparative and methodological context, we have also added broader syntheses on estuarine fish assemblages and on shoreline/intertidal fish sampling methods and their selectivity, and we now briefly refer to these works in the Introduction (when introducing ecological guilds of estuarine fish communities) and in the Methods/Discussion (when discussing the selectivity and limitations of the nearshore nets used in this study).
In addition, we have carefully checked all references to ensure consistency between in-text citations and the reference list (years, journal titles, volumes and page numbers), and we have verified that recent relevant studies on Yangtze estuary fish assemblages and sturgeon prey fields are included where appropriate. We believe that these revisions improve the completeness and coherence of the reference framework and better situate our local baseline study within the broader estuarine fish ecology literature.
Comments 7:I suggest changing the title to:
“Seasonal Variation of Shoreline Fish Assemblages at Two Stations in the Southern Branch of the Yangtze River Estuary”
This wording more accurately reflects the scope and strength of the dataset: it explicitly refers to shoreline assemblages, limits inference to two stations, and frames the work as an analysis of seasonal variation rather than a comprehensive description of all intertidal communities in the estuary. In this way, the title remains clear and informative while avoiding overstatement relative to the spatial and temporal coverage of the study.
Response 7:
We thank for this thoughtful suggestion. We agree that the proposed wording more accurately reflects the spatial scope and focus of our dataset. Accordingly, we have revised the title of the manuscript to: “Seasonal variation of shoreline fish assemblages at two stations in the southern branch of the Yangtze River Estuary”. This revised title explicitly indicates that the study concerns shoreline fish assemblages, limits inference to two stations along the southern branch, and emphasises the analysis of seasonal variation rather than a comprehensive description of all intertidal communities in the estuary.

Round 2
Reviewer 1 Report
Comments and Suggestions for Authors
After revision, the quality of this manuscript (Biology-2033955v2) has been significantly improved; I also agree with the authors' explanation of the revision. I have put forward some minor revision suggestions, hoping to be helpful for further enhancing the manuscript. I am not a native English speaker, so my comments will mainly not involve it.
Lines 21 and 42: The English name and Latin name of Acipenser sinensis are used interchangeably throughout the manuscript. It is recommended that both names be provided when the species is first mentioned in the abstract and main text, and a consistent single name be used for all subsequent references.
Line 83: The manuscript refers to "benthic and demersal fish" in multiple places, and it is suggested to retain only one. These two fish groups are not significantly different and difficult to distinguish clearly; moreover, the ecological classification adopted in the manuscript does not differentiate between them.
Lines 99 - 102: It is recommended to delete “using diversity-, similarity- and ABC-based metrics,” and rewrite the following one. Research objectives should not involve specific methods, and the next sentence (which addresses the research significance) should preferably be separated from the objectives into two independent sentences.
Line 135: The manuscript mentions "engineered shorelines" in multiple places, but I do not understand the rationale for emphasizing this term.
Line 191: The order of 6 - 10 Equations needs to be adjusted according to their order of appearance in the manuscript.
Lines 222 - 229: Section 2.3 can keep the second sentence and delete others, as the first “For all community analyses, ...” and third “Given the single net-set ...” sentences are redundant with previous content at lines 156 - 160. And use the abbreviation "ABC curve" as its full name has been mentioned earlier at the second sentence. Is it necessary to keep Section 2.3 as an independent section?
Line 230: The results section is described in detail but lacks conciseness. For example, the abbreviations of various indices have been mentioned earlier and can be used directly. Additionally, the textual description repeats all the results presented in the figures and tables, leading to redundancy.
Figure 4: The figure lacks a clear legend for “D, H’, C, and J’”.
Table 3: Capitalize the first letter for “fish” to maintain consistency with the headers of other figures/tables.
Lines 351 - 353: This conclusion requires stronger support. The low occurrence of A. sinensis in this area does not necessarily indicate that the contribution of this area is limited; it may be due to the inherently small population size of A. sinensis itself, and its occurrence in other water areas may be even rarer.
Lines 407 - 408: Capitalize the first letter of the sentence, and it is recommended that “pooled by season” be revised to “pooled to season analysis”.
Lines 426 - 430: In the conclusion section, sentences that merely restate specific results are unnecessary; it is recommended to refine or delete them.
Author Response
Comments 1: Lines 21 and 42: The English name and Latin name of Acipenser sinensis are used interchangeably throughout the manuscript. It is recommended that both names be provided when the species is first mentioned in the abstract and main text, and a consistent single name be used for all subsequent references. Response 1: Thank you for pointing this out. In the revised manuscript, we now provide both the English and Latin names when the species is first mentioned in the Abstract and in the Introduction (e.g., “Chinese sturgeon (Acipenser sinensis)”). For all subsequent references throughout the text, we use a single, consistent name (Chinese sturgeon / juvenile Chinese sturgeon), and we have removed the previous alternation between the English and Latin names. We have carefully checked the entire manuscript to ensure this consistency. Comments 2:Line 83: The manuscript refers to "benthic and demersal fish" in multiple places, and it is suggested to retain only one. These two fish groups are not significantly different and difficult to distinguish clearly; moreover, the ecological classification adopted in the manuscript does not differentiate between them. Response 2: Thank you for this helpful suggestion. In the revised manuscript, we have removed the repeated expression “benthic and demersal fish(es)” and now use a single, consistent term. Specifically, all such occurrences have been replaced by “demersal fishes” to match the ecological guild classification used in Table 2 and Figure 2. Comments 3:Lines 99 - 102: It is recommended to delete “using diversity-, similarity- and ABC-based metrics,” and rewrite the following one. Research objectives should not involve specific methods, and the next sentence (which addresses the research significance) should preferably be separated from the objectives into two independent sentences. Response 3: Thank you for this helpful suggestion. In the revised Introduction, we have removed the phrase “using diversity-, similarity- and ABC-based metrics” from the statement of research objectives. The objectives are now expressed without reference to specific analytical methods, and the subsequent sentence describing the research significance has been separated into an independent sentence. The revised text reads as follows: “This study therefore had three main objectives: (1) to document the taxonomic composition and ecological guild structure of shoreline fish assemblages in the southern branch of the Yangtze River Estuary; (2) to quantify their seasonal dynamics in terms of species richness, diversity and dominance across four sampling periods; and (3) to characterise inter-seasonal changes in community structure and similarity among seasons. Establishing these community-level patterns provides a standardized seasonal baseline for shoreline fish assemblages along this engineered estuarine sector and offers ecological background for future assessments of nursery and feeding habitats of juvenile A. sinensis.” Comments 4:Line 135: The manuscript mentions "engineered shorelines" in multiple places, but I do not understand the rationale for emphasizing this term. Response 4: Thank you for this comment. In the revised manuscript, we have removed the term “engineered shorelines” and replaced it with more neutral descriptions of the study area. Specifically, phrases such as “along an engineered shoreline (sector)” have been changed to “in the nearshore waters of the southern branch of the Yangtze River Estuary”, and in Section 2.1 the sites are now described simply as typical intertidal/nearshore waters of the southern branch. Because the precise coastal engineering classification is not essential to our objectives, we agree that using the more general term “nearshore waters” makes the manuscript clearer and avoids confusion. Comments 5:Line 191: The order of 6 - 10 Equations needs to be adjusted according to their order of appearance in the manuscript. Response 5: Thank you for pointing this out. In the revised manuscript, we have re-ordered and renumbered Equations (6)–(9) so that their numbering now follows the sequence in which they first appear in Section 2.2. All in-text references to these equations have been updated accordingly to ensure consistency. Comments 6:Lines 222 - 229: Section 2.3 can keep the second sentence and delete others, as the first “For all community analyses, ...” and third “Given the single net-set ...” sentences are redundant with previous content at lines 156 - 160. And use the abbreviation "ABC curve" as its full name has been mentioned earlier at the second sentence. Is it necessary to keep Section 2.3 as an independent section? Response 6: Thank you for this detailed suggestion. In the revised manuscript, we have removed the redundant first and third sentences from the former Section 2.3, as the pooling of catches by season and the limitation of a single net-set per station per season are already described earlier in the Methods and in the Discussion. The remaining sentence has been shortened and now refers directly to “ABC curves”, since the full term “abundance–biomass comparison (ABC) curves” is defined earlier. In addition, we have removed Section 2.3 as an independent subsection and incorporated this sentence into the end of Section 2.2 to avoid unnecessary fragmentation of the Methods. The revised sentence reads: “All statistical analyses were conducted in R (version 4.4), and ABC curves were generated using the ‘ggplot2’ package.” Comments 7:Line 230: The results section is described in detail but lacks conciseness. For example, the abbreviations of various indices have been mentioned earlier and can be used directly. Additionally, the textual description repeats all the results presented in the figures and tables, leading to redundancy. Response 7: Thank you for this constructive suggestion. In the revised manuscript, we have streamlined the Results section to improve conciseness. First, after defining each index by its full name and abbreviation in the Methods (e.g., Shannon-Wiener diversity index H′, Margalef richness D, Pielou evenness J′, Simpson index C), we now use only the abbreviations (H′, D, J′, C) throughout the Results to avoid repetition. Second, we have shortened the textual descriptions in Sections 3.X–3.X (diversity indices, similarity, alternation/migration indices and ABC curves) so that they highlight only the main patterns (ranges, seasonal maxima/minima and overall trends) rather than reiterating all numerical values already shown in the tables and figures. For example, the paragraph describing diversity indices now summarises the range of H′, D, J′ and C and identifies the seasons with the highest values, instead of listing each seasonal value in the text. We believe these changes reduce redundancy and make the Results more concise while preserving all essential information. Comments 8:Figure 4: The figure lacks a clear legend for “D, H’, C, and J’”. Response 8: Thank you for pointing this out. In the revised version of Figure 4, we have added a clear legend. Comments 9:Table 3: Capitalize the first letter for “fish” to maintain consistency with the headers of other figures/tables. Response 9: Thank you for this remark. In the revised manuscript, we have capitalized the first letter of “Fish” in the header/title of Table 3 to ensure consistency with the formatting of the other tables and figures. Comments 10:Lines 351 - 353: This conclusion requires stronger support. The low occurrence of A. sinensis in this area does not necessarily indicate that the contribution of this area is limited; it may be due to the inherently small population size of A. sinensis itself, and its occurrence in other water areas may be even rarer. Response 10: We thank the reviewer for this important clarification and fully agree with this point. In the revised Discussion, we have removed the previous statement implying that the contribution of this area to the population is limited and have softened our interpretation of the low encounter rate. We believe this revision better reflects the uncertainty associated with our limited spatial coverage and the overall depleted status of A. sinensis, and avoids overinterpreting the low encounter rate as direct evidence of a low contribution of this area to the population. Comments 11:Lines 407 - 408: Capitalize the first letter of the sentence, and it is recommended that “pooled by season” be revised to “pooled to season analysis”. Response 11: Thank you for this comment. In the revised manuscript, we have corrected the capitalization and rephrased this part for clarity. The sentence now reads: “Because samples from the two sites were pooled for seasonal analysis, potential small-scale differences between stations could not be evaluated.” We used the phrase “for seasonal analysis” here to express the intended meaning in grammatically correct English. Comments 12:Lines 426 - 430: In the conclusion section, sentences that merely restate specific results are unnecessary; it is recommended to refine or delete them. Response 12: Thank you for this helpful suggestion. In the revised manuscript, we have substantially condensed the Conclusion to avoid repeating detailed numerical results that are already presented in the Results and tables. We removed the sentences listing the exact ranges of diversity indices and the season-by-season numerical description of ABC outcomes, and we now focus on synthesising the main patterns and their implications. The Conclusion now emphasises (i) the dominant taxonomic and ecological characteristics of nearshore fish assemblages, (ii) the overall seasonal trends in diversity and disturbance, and (iii) the broader significance of these findings as a baseline for future monitoring and habitat assessments for juvenile A. sinensis, rather than restating specific numerical values.
